# CLUSTERING ENTITY SPECIFIC EMBEDDINGS TOWARDS A PRESCRIBED DISTRIBUTION

## ABSTRACT

Now ubiquitous in deep learning is the transformer architecture, which has advanced the state-of-the-art (SOTA) in a variety of disciplines. When employed with a bidirectional attention mask, a special [CLS] token is often appended to the sequence being processed, serving as a summary of the sequence as a whole once processed. While directly useful in many applications, the processed [CLS] embedding loses utility when asked to perform an entity-specific task given a multi-entity sequence - when processing a multi-speaker dialogue, for example, the [CLS] will describe the entire dialogue not a particular utterance. Existing approaches to address this often either involve redundant computation or non-trivial post-processing outside of the transformer. We propose a general method for deriving entity-specific embeddings *completely within* the transformer architecture, and demonstrate SOTA results in the domains of natural language processing (NLP) and sports analytics (SA), an exciting, relatively unexplored problem space. Furthermore, we propose a novel approach for deep-clustering towards a prescribed distribution in the absence of labels. Previous approaches towards distribution aware clustering required ground-truth labels, which are not always available. In addition to uncovering interesting signal in the domain of sport, we show how our distribution-aware clustering method yields new SOTA on the task of long-tail partial-label learning (LT-PLL). Code available upon publication.

## 1 INTRODUCTION

Now ubiquitous in deep learning is the transformer architecture, which has advanced the state-of-the-art (SOTA) in a variety of disciplines. Although the specifics vary by domain, at a high level a transformer model takes as input a set of embeddings $E$ and emits an *updated* set of embeddings $E'$ (Vaswani et al., 2017). When employed in the bi-directional setting, it is common to see a special [CLS] embedding appended to the sequence being analyzed, serving as a summary of the sequence as a whole once processed (Devlin et al., 2018; Dosovitskiy et al., 2020). While directly useful for performing many downstream tasks, the processed [CLS] embedding provides less utility for entity-specific tasks when given a multi-entity sequence. Consider the task of utterance-level emotion recognition in dialogue - the [CLS] embedding describes the dialogue, not any particular utterance.

Existing approaches towards deriving entity-specific embeddings are typically only able to construct embeddings for a single entity at a time, incurring redundant computation, or require non-trivial post-processing *outside* of the transformer architecture (§2.1). We posit that a combination of these factors is the reason the transformer architecture, which has advanced so many disparate fields, has not seen greater application in one of the most statistically-studied sports that is professional baseball. Baseball is described by a complex, non-discrete sequence of events influenced by multiple players, where each event is influenced by *at least* two players. This contrasts with the domain of language which is described by a discrete sequence of tokens, each influenced by only one entity.

The methods underpinning the majority of baseball statistics resemble the "bag-of-words" (*BoW*) approach from the early days of computational linguistics - a *human* expert identifies *important* in-game events, and players, teams, and managers are evaluated based on how many times they induce, or prevent, said events. Although this type of approach has provided benefits to the community, it also incurs the downsides of the BoW method, namely no ability for word-sense (event-sense) disambiguation. That is, current methods treat all of events of the same type as identical - a *single*

hit to deep left is considered the same as an infield *single*. However, these two different types of *singles* can have a different impact on the game which existing methods do not leverage.

To this end, we propose a novel, general approach for the derivation of entity-specific embeddings from a multi-entity sequence and apply the proposed method to the domains of NLP and sports analytics. By equipping a transformer-based model with our entity-specific tooling, we demonstrate appreciable gains in both the pre-training and fine-tuning phases of learning. In the domain of NLP, we demonstrate how our approach can be used to achieve SOTA results for the task of emotion recognition in conversation (ERC) on the MELD and EmoryNLP datasets. Furthermore, in the domain of sport, our approach outperforms current statistical approaches and a transformer-baseline.

Previous works in the realm of sport have sought to identify *clusters* of players who have similar short-term impact on the game, indicative of skill-level Heaton & Mitra (2022). To this end, we propose a method for distribution-aware clustering in the absence of labels. Consider the histogram of starting pitcher Wins Above Replacement (WAR) in the MLB from 2015-2021, presented in Figure 1. A proxy for player skill-level, it is clear that WAR is not evenly distributed across the player population. By taking advantage of this population-level heuristic, we can guide the model to construct more informative player *form* clusters.

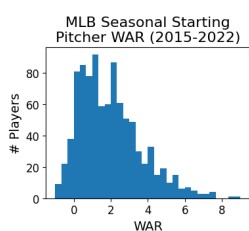

Figure 1: Pitcher WAR.

Existing methods for distribution-aware deep-clustering have found success when applied to the task of long-tail partial-label learning (LT-PLL) (§2.2). Such approaches frame classification as a clustering task, with emphasis on improving performance on the "long-tail" of infrequent labels. However, these methods require the presence of ground-truth labels to account for the poor-calibration of the cluster-assignment mechanism in the early stages of training. Applying our distribution-aware clustering methodology to the task of LT-PLL yields new SOTA performance with lower variance.

Our contributions are as follows:

- Propose a novel method for deriving entity-specific embeddings from a multi-entity sequence which achieves SOTA performance in ERC and player performance projection

- Present a method for distribution-aware clustering in the absence of labels, and demonstrate new SOTA on the task of LT-PLL

- Demonstrate the utility of applying modern representation learning techniques to the domain of sport, identifying a discrete notion of player *form* and highlighting how the domain of sport can offer new perspectives on age-old topics in machine learning such as clustering

## 2 RELATED WORK

### 2.1 ENTITY-SPECIFIC EMBEDDINGS FROM MULTI-ENTITY SEQUENCES

Multi-entity sequences appear often in the realm of NLP in the form of multi-speaker dialogues, where the "entity" may be an individual speaker or utterance (Zhang et al., 2019; Wang et al., 2011). Emotion recognition in conversation (ERC) is a common task to perform on such data, with three popular benchmark datasets being MELD (Poria et al., 2018), EmoryNLP (Zahiri & Choi, 2017), and IEMOCAP (Busso et al., 2008). In analyzing the leaderboard[1] for these benchmarks, with focus on text-based approaches, two approaches for isolating entity-specific signal emerge - 1) training the model to saturate the [CLS] embedding with signal directed towards a particular segment of the dialogue, or 2) non-trivial post-processing of the sequence *outside* of the transformer.

The SPCL-CL-ERC model is an example of the former, currently SOTA on MELD and EmoryNLP (Song et al., 2022a). Given a sequence of utterances $U = [u_1, u_2, ..., u_t]$ and corresponding speakers $S = [s_1, s_2, ..., s_t]$, Song et al. format the input sequence as $X_i = [s_1, u_1, ..., s_{t-1}, u_{t-1}, s_t, u_t, Q]$ where Q is constructed as "for $u_t$, $s_t$ feels [MASK]". Son et al. propose a similar prompting-based method that can classify individual utterances or the relationship between two utterances (Son et al., 2022). InstructERC (Lei et al., 2023) applies a prompting+retrieval technique on autoregressive large language models, specifically Llama2 (Touvron et al., 2023). While these methods have been

---

[1]https://paperswithcode.com/task/emotion-recognition-in-conversation

fruitful in their application, they incur a significant amount of redundant computation. If a dialogue $D$ contains N utterances, $D$ must be processed $N$ times to make a prediction for each utterance.

The EmotionIC model is an example of an approach that requires non-trivial post-processing outside the core transformer architecture (Liu et al., 2023b). A pre-trained RoBERTa model (Liu et al., 2019) first processes the sequence of words comprising the dialogue, and speaker-/utterance-level features are extracted by post-processing via RNN modules. The idea of having a pre-trained language model "backbone" process the textual sequence before a supplemental module(s) isolates the signal of interest has been well-explored (Huang et al., 2019; Shmueli & Ku, 2019; Song et al., 2022b). A first-step in many of the post-processing approaches is the *averaging* of context embeddings that correspond with a particular utterance/speaker. We speculate that this could be improved, as Reimers & Gurevych (2019) have demonstrated that a simple average of processed contextual embeddings provides less utility than a learned function of the same. HiDialog (Liu et al., 2023a) derives entity embeddings from RoBERTa without the use of averaging, but still requires the post-processing of said embeddings by a graph neural network (GNN). Given transformers are a special case of GNN (Veličković, 2023), we posit this operation can be performed *inside* the model.

## 2.2 DEEP-CLUSTERING

Deep-clustering has been used as a means to an end, *i.e.* as a pre-training objective for representation learning models (Caron et al., 2018; Asano et al., 2019), as well as the downstream task a model is ultimately trained to perform (Wang et al., 2022; Tsitsulin et al., 2020). A key aspect of the modeling pipeline is the cluster-assignment module, often the Sinkhorn-Knopp algorithm Sinkhorn (1967).

Given a matrix $Q \in \mathcal{R}^{NxK}$ as input the iterative Sinkhorn-Knopp algorithm returns an updated matrix $Q'$ in which all rows sum to one and the sum of each column is roughly equal (Asano et al., 2019). This is achieved by iteratively dividing $Q$ by its column- and row-sum. Interpreted through the lens of clustering, given a raw set of cluster *predictions* $Q$, the Sinkhorn-Knopp algorithm yields a set of cluster *assignments* $Q'$ in which cluster assignments are *evenly* distributed and each record has a 100% probability of being assigned to one of the clusters.

The assumption of uniform cluster distribution yields good results when leveraged for pre-training, but is counter-intuitive in real-world scenarios, where labels are *seldom* evenly distributed. To address this, Wang et al. (2022) propose the *Sinkhorn Label Refinery (SoLar)*, a modified version of the Sinkhorn-Knopp algorithm that yields cluster assignments aligned with a *prescribed* distribution, intended for the task of long-tail partial-label learning (LT-PLL). Specifically, SoLar seeks probability assignments $Q = \left\{ \mathbf{Q} \in \mathbb{R}_+^{N \times L} | \mathbf{Q}^T \mathbf{1}_n = \mathbf{r}, \mathbf{Q} \mathbf{1}_L = \mathbf{c}, q_{ij} = 0 \text{ if } j \notin S_i \right\}$ where $\mathbf{r}$ is the ground-truth label distribution, $L$ the number of classes, $\mathbf{c} = \frac{1}{n} \mathbf{1}_n$, and $S_i$ the set of candidate labels for record $i$, relevant for the PLL task. In practice, label assignments are obtained by computing $\mathbf{Q} = n \cdot \text{diag}(\boldsymbol{\alpha}) \mathbf{M} \text{diag}(\boldsymbol{\beta})$, where $\mathbf{M}$ is the matrix containing *raw* cluster membership predictions and $\boldsymbol{\alpha}$ and $\boldsymbol{\beta}$ are iteratively updated as $\boldsymbol{\alpha} \leftarrow \mathbf{c}/(\mathbf{M}\boldsymbol{\beta})$ and $\boldsymbol{\beta} \leftarrow \mathbf{r}/(\mathbf{M}^T \boldsymbol{\alpha})$.

While SoLar established new SOTA performance on the CIFAR10-LT and CIFAR100-LT benchmarks, it requires ground-truth labels to yield cluster assignments aligned with a prescribed distribution. As the authors note, SoLar yields poorly aligned cluster assignments in the early stages of training when the model emits low-quality predictions. To counteract this, the ground-truth label associated with each record is leveraged to identify records that assigned to the *correct* cluster with high confidence. Without the ground-truth labels, SoLar struggles to yield accurate cluster assignments aligned with a prescribed distribution, limiting its application to the supervised setting.

## 2.3 PLAYER PROJECTION IN PROFESSIONAL BASEBALL

In this section we briefly describe the current approaches towards player performance projection in professional baseball. Elfrink (2018), Alceo & Henriques (2020), and Valero (2016) explore how machine learning (*i.e.* linear regression, logistic regression, XGBoost, and support vector machines) can be used to make predictions about the game of baseball. Input features to their models were simple counting stastistics or higher-order statistics derived therefrom. One example, on base percentage (OBP) is computed as $OBP = \frac{H+BB+HBP}{AB+BB+SF+HBP}$ where $H$, $BB$, $HBP$, $AB$, and $SF$ are the *number of times* a player participated in or induced a *hit*, *walk*, *hit by pitch*, *at-bat*, and *sacrifice*

*fly*, respectively. We direct the curious reader to the original papers for an exhaustive list, but at a high level, these higher-order statistics are computed by manipulating the counting statistics in the numerator and denominator to try and isolate signal describing a particular aspect of a player. The method to compute WAR (§1) is perhaps more sophisticated, but is still ultimately derived from simple counting statistics, which we view as a rather crude description of the game.

Silver & Huffman (2021) and Sun et al. (2023) explore how predictions in this domain can be improved by leveraging deep learning (DL) architectures of MLP and LSTM. However, the features used as input to these models are largely the same as described above - counting statistics or metrics derived therefrom. While these DL models learn to construct internal representations of players, such representations can only be created from the simple counting statistics which they are given. Seeing how traditional disciplines of DL have long since moved on from hand-crafted features, we believe there is large potential to adapt these findings to the domain of sport, particularly baseball.

## 3 OUR APPROACH

### 3.1 DERIVING ENTITY SPECIFIC EMBEDDINGS

Our approach for deriving entity-specific embeddings begins by introducing a new [ENTITY] embedding in to the model's vocabulary. Records are first processed as they would be normally - *i.e.* vanilla construction of the input sequence, attention mask, padding mask, etc - before any entity-specific modifications. Entity-specific implementation begins by identifying the entities $E = \{e_1, e_2, ..., e_N\}$ for which embeddings should be derived. We assume no strict definition of *entity*, using it in the basic sense to simply denote sub-sections of the context which are of interest.

Once the entities are identified, a boolean mask denoting portions of the input sequence with with each entity interacts $T = \{t_{e_1}, t_{e_2}, ..., t_{e_N}\}$, where $t_{e_i} \in [0, 1]^{1 \times SeqLen}$, is constructed. This is often as simple as identifying the speaker or utterance ID associated with each timestep of the sequence. Then, $|E|$ entity embeddings are appended to the input sequence, and $T$ is used to update the attention mask such that embedding $e_i$ can only tend to indices in which entity $i$ participates. Each entity embedding is also allowed to tend to itself and the [CLS] embedding.

At a high level, the proposed entity embeddings can be seen as a *generalization* of the vanilla [CLS] embedding, which absorbs information from the entire record useful in performing record-level tasks. Our entity embedding serves a similar role, directed at a particular subset of the context. in pre-training, the entity embeddings can be used to better perform the pre-training task on indices in which each entity participates. In fine-tuning, the entity embeddings can be adapted to specific ends.

In some applications, such as ERC, it may be useful to leverage the relationship *between* the entities in a given sequence - utterance $i$ may influence utterance $i + 1$. Although this has been performed *outside* the transformer, often via a GNN (§2.1), a transformer can be viewed as a special type of GNN, leading us to posit that this post-processing can be done equally as well *within* the transformer. To this end, we explore how the attention mask described above can be *opened up* to allow entities to tend to one another, dubbing this "entity-to-entity" (E2E) attention. In doing so, a set of learnable position embeddings is applied to the entity embeddings, instilling a temporal order therein. An example of our approach with E2E enabled is presented in Figure 2, with E2E depicted in *yellow*. When E2E is not enabled, the cells depicted in yellow are closed (colored *white*).

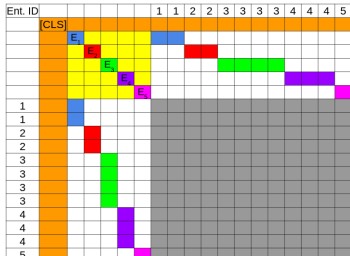

Figure 2: Example attention mask for our approach. *Ent. ID* included for illustration purposes only. E2E attention denoted in yellow.

### 3.2 CLUSTERING TOWARDS A PRESCRIBED DISTRIBUTION

The approach described above will equip a model with the tooling to derive entity-specific embeddings from a multi-entity sequence, but we are curious how these embeddings can be constructed towards specific ends. Heaton & Mitra (2022) propose a two-step method for identifying player *form* - their *short-term* skill-level - in professional baseball. In stage one a representation learning model

learned to describe a player in the short term, and in stage two these representations were clustered to obtain a discrete notion of player *form*. Given that skill is not evenly distributed among the player population (Figure 1), we explore how an end-to-end deep learning model can be encouraged to emit cluster assignments aligned with a *prescribed* distribution.

Specifically, we seek a modified version of the Sinkhorn Knopp algorithm that is encouraged to assign a specific number of records to each cluster, in accordance to a prescribed distribution. That is, we seek a matrix $Q = \left\{ \boldsymbol{Q} \in \mathbb{R}_+^{K \times N} | \boldsymbol{Q}\boldsymbol{1} = \boldsymbol{\beta}, \boldsymbol{Q}^T \boldsymbol{1} = \frac{1}{N} \cdot \boldsymbol{1} \right\}$ where the vector $\boldsymbol{\beta}$ denotes the desired number of members in each cluster. We note that this is equivalent to the restrictions placed on the Q matrix in *SoLar* without the LT-PLL constraints (§2.2), but the process to arrive at Q differs.

Instead of iteratively dividing raw predictions $\mathbf{M} \in \mathbb{R}_+^{K \times N}$ by the column and row sum, as in the vanilla Sinkhorn-Knopp implementation, we iteratively divide by the column sum and *scaled* row sum to control the amount of probability mass that is allowed to accumulate over each cluster. Computed as scaled row sum $= \frac{\text{row sum}}{\beta \cdot N}$, the *scaled* row sum denotes the prescribed cluster *population* within each batch of predictions. The vanilla Sinkhorn Knopp algorithm can be recovered by simply setting $\boldsymbol{\beta} = \frac{\boldsymbol{1_k}}{k}$, where $\boldsymbol{1_k}$ denotes a vector of ones with shape $k$. Python-like pseudocode for our *Scaled Sinkhorn-Knopp* algorithm is presented in §A.1.1.

# 4 EXPERIMENTS

## 4.1 DERIVING ENTITY-SPECIFIC EMBEDDINGS

### 4.1.1 EMOTION RECOGNITION IN CONVERSATION

Application begins by adding an [ENTITY] embedding to the vocabulary of the pre-trained `RoBERTa-large` model, initialized as the learned [CLS] embedding. Both tokens serve a similar role - absorbing information describing (a portion of) the input sequence - so the [CLS] embedding serves as a good starting point. For ERC, an entity-embedding is constructed for each *utterance* in the dialogue, and we enable *entity-to-entity* (*E2E*) attention to allow the model to leverage the relationship *between* each utterance in the dialogue. Additive positional embeddings applied to the entity embeddings are initialized as the pre-trained positional embeddings, but optimized separately.

The modified model is first subjected to a regime of *extended pre-training* so it can learn to use the new parameters before being applied to the task of ERC on the datasets described below. During pre-training the model performs MLM and we gauge model performance in terms of *uncertainty*, defined as $U(\mathcal{P}) = \frac{1}{N} \sum_{i=1}^{N} \frac{1}{p_i}$ where $p_i$ is the probability the model emitted for the correct label for prediction $i$. Similar to perplexity, lower scores are better. In fine-tuning, the model minimizes a combination of cross-entropy and prototype-cosine similarity loss, weighted 9:1. The prototype for each label is constructed by randomly sampling 64 of the 256 most-recently processed records with that label and taking the average. Model is evaluated in terms of weighted F1 score and compared against EmotionIC, HiDialog, SPCL-CL-ERC, and InstructERC (§2.1).

### 4.1.2 PLAYER PERFORMANCE PROJECTION

We apply the method to the task of player performance projection in the MLB, highlighting the general nature of the approach. A model is trained from scratch as no pre-trained models exist.

We devise a infilling-based pre-training scheme very similar to RoBERTa's *masked language modeling* (MLM) which we term *masked game modeling* (MGM). When presented with a sequence of events describing $N$ games, roughly 15% of the timesteps are masked and the model is tasked with predicting what happened on the field at that point in time. During this process, the model will learn to saturate the entity embeddings with signal useful in discerning how a particular player impacts the game. Then, this model can be fine-tuned to predict how players will perform in a particular game given their last 15 games. Specifically, the model predicts how many strikeouts (K), hits (H), and walks (BB) starting pitchers will record against the opposing team's starting batters and vice-versa.

We train an XGBoost model to make the same predictions using traditional statistics describing the player's performance in the last 15 games. A mutual-information based feature selection strategy was used to identify the input features most useful for performing each task with feature counts

of 10, 25, 50, and 100 explored. We also compare against a transformer-baseline in which entity embeddings are constructed by extracting and *averaging* the processed contextual embeddings corresponding to each entity, similar to how *entity* embeddings are often derived in NLP (§2.1).

## 4.2 CLUSTERING TOWARDS A PRESCRIBED DISTRIBUTION

### 4.2.1 PRESCRIBED VS ASSIGNED DISTRIBUTION

As mentioned by the authors, *SoLar* will yield mis-aligned cluster assignments in the early stages of training, *i.e.* when the model emits poor cluster predictions (Wang et al., 2022). We explore how the assignments from *SoLar* and our *Scaled Sinkhorn-Knopp* assignemnt mechanism differ in the face of low-quality predictions. If the cluster *assignments* do not align with the prescribed distribution, then it would be unreasonable to expect the *predicted* distribution to be aligned.

We explore how the initial cluster predictions influence the results of each assignment mechanism by simulating predictions of varying quality. Given a prescribed probability distribution $\beta = \{\beta_0, \beta_1, ..., \beta_k\}$, we construct a set of synthetic predictions $P$ of shape $k \times N$ where $N * \beta_k$ records will have a probability $\mu$ of being assigned to cluster $k$. By varying the $\mu$ parameter we can simulate predictions of varying quality, higher $\mu$ values indicating a higher quality model. $\mu = -1$ is used to denote random predictions. We create 500 batches of predictions $P$, subjecting them to each of the cluster assignment mechanisms, and quantify the degree to which the assigned distribution aligns with the prescribed distribution in terms of KL divergence, $D_{KL}$ Kullback & Leibler (1951).

### 4.2.2 LEARNING PLAYER FORM

Inspired by Heaton & Mitra (2022), we explore how a notion of player *form* - a description of short-term performance - can be uncovered by an end-to-end clustering model. While the player embeddings constructed in §4.1.2 contain signal useful for predicting player performance, we explore how the model can be used to identify *clusters* of players who impact the game in a similar fashion. This set of experiments employ a four-phase training scheme that utilizes three different losses. The first objective used in training is MGM, as described in §4.1.2. MGM is complemented by two contrastive learning objectives, *MoCo v3* (Chen et al., 2021) and *swapped clustering* (Caron et al., 2020), which encourage the model to yeild *similar* representations for *similar* inputs. We briefly describe these objectives, but direct the curious reader to the original papers.

Two models are maintained during training, the main, *query* model, $f_q$ and a supplemental, *key* model, $f_k$, where $f_k$ is an exponential moving average of $f_q$. $f_k$ is used *only* towards the *MoCo v3* loss. When presented with two sequences of events describing the same player at two close points in time, $x_1$ and $x_2$, both $f_q$ and $f_k$ will emit a set of player embeddings, $q_1 = f_q(x_1)$, $q_2 = f_q(x_2)$, $k_1 = f_k(x_1)$ and $k_2 = f_k(x_2)$. To minimize the *MoCo v3* loss, the model must learn to identify $k_2$ from the set of all momentum-encoded records given $q_1$, and $k_1$ given $q_2$. For *swapped clustering*, initial cluster predictions are made for $q_1$ and $q_2$ by computing their cosine similarity with $K$ cluster prototypes $\{c_1, ...c_k\}$. Cluster *assignments* are then made by subjecting the raw predictions to our Scaled Sinkhorn-Knopp algorithm (§3.2), yielding cluster assignments that are aligned with a prescribed distribution. To minimize this loss, the model must learn to make cluster *predictions* that are aligned with the *Scaled Sinkhorn-Knopp* assignments.

The overall loss is $\mathcal{L} = w_{MGM} \cdot \mathcal{L}_{MGM} + w_{MoCo} \cdot \mathcal{L}_{MoCo} + w_{Cluster} \cdot \mathcal{L}_{Cluster}$ where $w_{MGM}$, $w_{MoCo}$, and $w_{Cluster}$ are the weights associated with each objective and changed in each phase of training, described in Table 1. Cluster quality will quantified in terms of KL Divergence against the *prescribed* distribution and cosine-distance Silhouette score Rousseeuw (1987).

| Phase | MGM | MoCo | Cluster |
|-------|-----|------|---------|
| 1 | 1.0 | 0.0 | 0.0 |
| 2 | 0.5 | 0.5 | 0.0 |
| 3 | 0.5 | 0.5→0.25 | 0.0→0.25 |
| 4 | 0.5 | 0.25 | 0.25 |

Table 1: Objective schedule.

### 4.2.3 LT-PLL

We apply our *Scaled Sinkhorn-Knopp* to the LT-PLL task, framing classification as a clustering task. Here, each record $x_i \in X$ has a candidate label set $s_i \in Y$, where $Y = 1, ..., k$ is the label space, and the objective is to identify the *ground-truth* label from the candidate set. Specifically, we adopt the pipeline proposed by Wang et al. (2022), the current cluster-based SOTA on this task, and simply use our *Scaled Sinkhorn-Knopp* cluster assignment mechanism in place of their *SoLar*

assignment mechanism - everything else is preserved. The approach sees an 18-layer ResNet trained for 1,000 epochs using a SGD optimizer. We are primarily concerned with the impact our *Scaled Sinkhorn Knopp* cluster assignment mechanism has on performance, and direct the interested reader to the original paper for complete pipeline details. We also perform experiments with RECORDS debiasing applied to our modified *SoLar* pipeline (Hong et al., 2023).

### 4.3 DATASETS

We identify three multi-speaker dialogue datasets for use in our ERC experiments, MELD (Poria et al., 2018), EmoryNLP (Zahiri & Choi, 2017), and IEMOCAP (Busso et al., 2008). MELD and EmoryNLP have prescribed train/test splits which are used when appropriate for pre-training and fine-tuning, but IEMOCAP does not and thus only used in pre-training. EmoryNLP contains 12,606 utterances from 897 dialogues, MELD 13,708 utterances from 1,433 dialogues, and IEMOCAP 7,433 utterances from 151 dialogues. Utterances from MELD/EmoryNLP are labeled as expressing one of seven emotions, while IEMOCAP contains 10 possible labels, including *other*.

As in (Wang et al., 2022), we explore the task of LT-PLL on the CIFAR10-LT and CIFAR100-LT datasets (Cao et al., 2019; Wei et al., 2021). The original CIFAR-10 and CIFAR-100 datasets contain roughly 50,000/10,000 training/test images labeled with 10 and 100 classes, respectively. In the "long-tail" (LT) version, an imbalance ratio $\gamma$ denotes the ratio of population in the most abundant to least abundant class, with an exponential decay between.

We employ the same methodology described by Heaton & Mitra (2022) to construct our professional baseball dataset, expanded to included data through the 2021 MLB season. The resulting dataset contains 4.6 million pitches thrown in 16k different games. This tabular data is converted to a pitch-by-pitch sequence describing the 1) gamestate when each pitch was thrown, 2) the pitcher and batter involved, and 3) change in gamestate resulting from the pitch via real-valued statistics and learned embeddings. Data is presented to the model the same as in Heaton & Mitra (2022).

## 5 RESULTS

### 5.1 DERIVING ENTITY-SPECIFIC EMBEDDINGS

#### 5.1.1 PRE-TRAINING

Presented in Tables 2 & 3 are metrics describing the impact of our entity embedding method during pre-training in the realm of language and sport, respectively. For sake of space, training parameters are presented in §A.2. Analyzing each set of results leads to similar conclusions. Our entity-embedding approach improves pre-training performance by lowering the uncertainty score by 2.59% in the domain of language and 71.55% in the domain of sport. Although strong improvements in both domains, we suspect the improvement on the ladder is because the model was equipped with the entity embedding tooling *from the start*. If pre-trained on a larger corpus, we hypothesize the improvements in NLP would increase in magnitude.

| Model | Entity Embds | Fine-tune | Unc. |
|---|---|---|---|
| RoBERTa | | | 2.79 |
| RoBERTa | | ✓ | 2.32 |
| RoBERTa | ✓ | | 2.58 |
| RoBERTa | ✓ | ✓ | **2.26** |

Table 2: Uncertainty scores (Unc, §4.1.1) from performing extended pre-training.

| Config | Unc. | F1 |
|---|---|---|
| Vanilla | 20.14 | 0.55 |
| Entity Embeddings | **5.73** | **0.63** |

Table 3: Performance results for MGM after of pre-training on sequences of in-game events from 2015-2020. Metrics are uncertainty (Unc,§4.1.1) and weighted F1-score (F1).

#### 5.1.2 FINE-TUNING

We then applied the (extended-)pre-trained models to the tasks of ERC and player performance projection, presenting the results in Tables 4 and 5, respectively. Full training details are presented in §A.3. Predictions evaluated on mean squared error (MSE) and $R^2$ (Wright, 1921).

These results suggest that the improved pre-training performance translates to improved performance on downstream tasks from two separate domains. In the NLP task of ERC, our model establishes new SOTA performance on the EmoryNLP dataset, and when ensembled with the SPCL-CL-

ERC a new SOTA on MELD. Our approach is also competitive with InstructERC, based on the much larger Llama2 model (350M vs 7B parms). Models equipped with our entity embedding method also outperform our two baselines for the task of player performance projection in the MLB. Only in a small number of cases do the baselines yield stronger performance than our approach, and all of the cases are in terms of the MSE metric. Upon inspection, the baseline emits predictions clustered around the global average, leading our method to be preferred. These results demonstrate advantages of using representation learning techniques in place of the typical statistics-based, *BoW* approach.

| Model | MELD | EmoryNLP |
|---|---|---|
| EmotionIC | 66.40 | 40.01 |
| HiDialog | 66.96 | N/A |
| SPCL-CL-ERC | 67.25 | 40.94 |
| InstructERC | 69.15 | 41.39 |
| EE (Ours) | 66.53 | 40.69 |
| EE+E2E (Ours) | 66.91 | 41.98 |
| EE+E2E (Ours) & SPCL-CL-ERC | **70.26** | **47.61** |

Table 4: Fine-tuning scores (Weighted-F1). EE: entity embeddings, E2E: entity-to-entity attention.

| Config | Strikeouts (K) | | | | Walks (BB) | | | | Hits (H) | | | | Average | |
|---|---|---|---|---|---|---|---|---|---|---|---|---|---|---|
| | Pitcher | | Batter | | Pitcher | | Batter | | Pitcher | | Batter | | | |
| | MSE | R² | MSE | R² | MSE | R² | MSE | R² | MSE | R² | MSE | R² | MSE | R² |
| Stat | 5.43 | 0.20 | **0.43** | 0.05 | 1.73 | 0.04 | 0.19 | **0.02** | 4.48 | 0.07 | **0.43** | 0.03 | 2.11 | 0.07 |
| Embd Avg | 5.46 | 0.20 | 0.44 | 0.04 | 1.49 | 0.03 | **0.17** | 0.01 | 4.71 | 0.06 | **0.43** | 0.03 | 2.12 | 0.06 |
| Entity Embds | **5.34** | **0.22** | **0.43** | **0.06** | **1.48** | **0.05** | **0.17** | **0.02** | 4.57 | **0.08** | **0.43** | **0.04** | **2.07** | **0.08** |

Table 5: Fine-tuning results. Metrics are Mean Squared Error (MSE) and (R²).

## 5.2 CLUSTERING TOWARDS A PRESCRIBED DISTRIBUTION

### 5.2.1 PRESCRIBED VS ASSIGNED DISTRIBUTION

Presented in Figure 3a is the KL-Divergence between the *prescribed* cluster distribution and the distribution of cluster *assignments* from *SoLar* and our *Scaled Sinkhorn-Knopp*. We observe that both assignment mechanisms yield more aligned assignments as the raw prediction quality improves (*i.e.* $\mu \uparrow$), but that our *Scaled Sinkhorn-Knopp* yields significantly more aligned assignments in the face of particularly low-quality predictions. Inspection of the resulting assignments (§A.4) demonstrates *SoLar* tends to *overestimate/underestimate* the most/least populous classes given poor predictions.

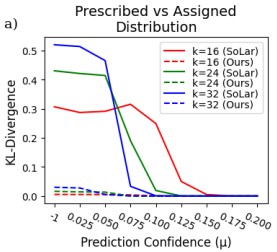 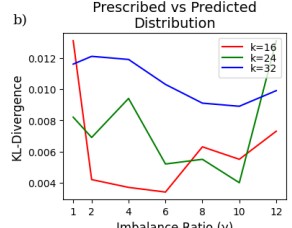 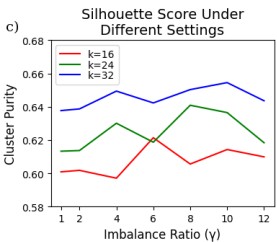

Figure 3: a) KL-Divergence in prescribed and assigned distribution from *SoLar* and *our* assignment mechanism b) KL-Divergence in prescribe vs resulting distribution and c) Cluster silhouette scores.

### 5.2.2 LEARNING PLAYER FORM

Presented in Figures 3b and 3c are results of applying our distribution-aware clustering method to representations of *starting pitchers* in the MLB. Figure 3b describes how well the model's *predicted* cluster distribution aligns with the *prescribed* cluster distribution under different configurations of cluster count ($k$) and imbalance ratio ($\gamma$), and 3c describes how those same configurations translate to cluster quality. We see that the KL-Divergence is not minimized when the model is encouraged to emit *uniformly-distributed* cluster assignments (*i.e.* $\gamma = 1$), but when the prescribed distribution is *imbalanced*. This is reinforced by Figure 3c, which describes the cosine distance silhouette scores from the same configurations. For all choices of $k$, we see that the silhouette score is not maximized when the model is encouraged to find uniformly-distributed clusters, but towards some prescribed, *skewed* distribution. The value of $\gamma$ that yields the highest silhouette score varies based on choice of $k$, but we see that silhouette score is often *maximized* when the KL-Divergence is *minimized*. These results reinforce the notion that the underlying records themselves are not uniformly distributed.

We then perform `norm_sort` (§A.1.2) on the cluster assignments with the highest silhouette score ($k = 32$, $\gamma = 10$), sorting clusters based on population size and then visualizing how this corresponds with cluster WAR, presenting results in Figure 4. For each player, we identify their WAR for each season in which they played and to which *form* clusters they were assigned during the same year, computing the *percent of the season* they spent in each form. Then, multiply the form-distribution vector by their seasonal WAR to identify the WAR associated with each cluster. The total, population-wide WAR attributed to each cluster is then divided by the average percent of each season spent in each cluster, yielding a season-long WAR value per cluster.

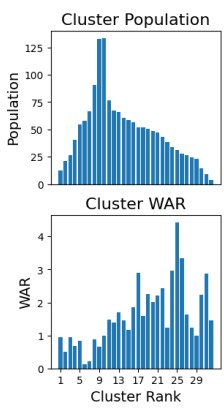

These findings closely mirror Figure 1 - there is a large population of *average* players with dwindling populations to either end. Although correlated with WAR, a proxy for player quality, anomalies do exist - some low-WAR players are assigned to high-WAR clusters and vice-versa. We posit this presents potential *arbitrage* opportunities for professional teams - that is, finding players who are highly rated by our representation learning model, but poorly-rated in terms of traditional statistics. This idea was the basis for the original statistical revolution in baseball, where counting statistics were used to find players who were under-valued by human perception.

Figure 4: Norm-sorted cluster pop. & WAR.

### 5.2.3 LT-PLL

Here we present the results of performing the experiments for LT-PLL. Five trials were performed for each experiment, with the mean and standard deviation reported. We use the codebase provided by Wang et al. (2022) swapping the cluster assignment mechanism and adding RECORDS debiasing. Performance on CIFAR10-LT/CIFAR100-LT with flipping probability $\phi = 0.5/0.1$ and imbalance ratio $\gamma = 100/20$, respectively, are presented in Table 6. Performance is evaluated on the population level as well on different groups of label classes with varying population (*Many/Few/Medium*). *Many* denotes the 33% *most* populous label classes, *Few* the 33% *least* populous, and *Medium* the rest.

| Method | CIFAR10-LT | | | | CIFAR100-LT | | | |
|---|---|---|---|---|---|---|---|---|
| | All | Many | Medium | Few | All | Many | Medium | Few |
| SoLar (reported) | 76.64 | 96.50 | 76.01 | 49.34 | 53.03 | 74.33 | 54.09 | 30.62 |
| SoLar (reproduced) | 74.90±2.37 | **96.26±0.60** | 74.73±3.55 | 53.58±4.14 | 52.56±1.30 | **76.43±0.78** | 53.21±1.77 | 28.03±3.13 |
| Ours | 75.72±0.79 | 95.87±0.45 | 75.57±1.30 | 55.79±1.45 | 52.69±0.80 | 75.92±0.56 | 53.70±0.47 | 28.42±2.22 |
| Ours + RECORDS | **80.95±1.39** | 89.81±1.30 | **78.72±2.71** | **75.06±3.21** | **54.05±0.90** | 74.32±0.26 | **56.40±0.28** | **31.35±2.73** |

Table 6: Accuracy on CIFAR10-LT/CIFAR100-LT with $\phi = 0.5/0.1$ and $\gamma = 100/20$, respectively.

Simply replacing the Sinkhorn-Knopp cluster assignment from *SoLar* with our *Scaled Sinkhorn-Knopp* yields higher average accuracy with lower variance across five trials. Performance is further improved by applying RECORDS debiasing. While the performance of our model is slightly lower than SoLar in identifying the most abundant class (*Many*), our model is more accurate on the "long tail" of labels (*Medium/Few*), which are the classes of highest interest. Our method also outperforms CORR+RECORDS (Hong et al., 2023) in a suite of experiments presented in A.5.

## 6 CONCLUSION

Above we have introduced our novel approach for deriving entity-specific embeddings from a multi-entity sequence, and how said embeddings can be clustered towards a prescribed distribution in the absence of labels. Models equipped with our entity-embedding tooling demonstrate superior performance in both pre-training and fine-tuning compared to vanilla counterparts, leading to new SOTA in ERC and player performance projection. Our distribution-aware clustering method can be used to uncover similar groups of players in the realm of sport and to establish new SOTA on the task of LT-PLL. We hope these findings will demonstrate how the realm of sport is an interesting, relatively un-explored problem space that can lead to new perspectives on age-old topics.

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

# A APPENDIX

## A.1 ALGORITHMS

### A.1.1 SCALED SINKHORN-KNOPP

**Algorithm 1** Scaled Sinkhorn-Knopp Pseudocode

**Inputs:** KxB robability matrix Q, desired distribution $\beta$
**Outputs:** KxB Probability matrix Q aligned to desired distribution

```
def scaled_sinkhorn_knopp(Q, beta, n_iters=25):
    bsz = Q.shape[1]
    beta *= (bsz / beta.sum())          # sum(beta) = bsz
    for it in range(n_iters):
        row_sum = Q.sum(dim=1)
        scaled_sum = row_sum / beta     # scale by prescribed population
        Q = Q / scaled_sum / bsz
        col_sum = Q.sum(dim=0)
        Q = Q / col_sum / bsz
    return Q*bsz
```

### A.1.2 NORM_SORT

## A.2 PRE-TRAINING PARAMETERS

### A.2.1 NLP

Metrics presented in Table 2 describe a RoBERTa model subjected to 15 epochs extented pre-training on the MELD, EmoryNLP, and IEMOCAP datasets with an Adam optimizer, batch size of 32, $l2$ weight of 0.01, a learning rate of $1e - 5$ with a cosine schedule and 150 iterations of warmup.

### A.2.2 SPORT

Metrics presented in Table 3 describe an 8 layer, pre-norm transformer model Xiong et al. (2020) with an internal dimension of 768 and 8 attention heads was trained using an Adam optimizer with a learning rate of `5e-4`, *L2* weight of `1e-4`, and a batch sizes of 30 on 2.5M records.

---

**Algorithm 2** Normal Sort

---

**Inputs:** List of cluster population sizes P, list of cluster values V
**Outputs:** Sorted P and W such that P approximates a normal distribution with V encouraged to be in increasing order.

---

```python
def normal_sort(P, V):
    # sort by population, largest to smallest
    agg = list(sorted(zip(P, V), key=lambda x: x[0], reverse=True))
    ordered = []
    sentinel_value = None

    for pop_size, val in agg:
        if len(ordered) == 0:
            sentinel_value = val
            ordered.append([pop_size, val])
        elif val < sentinel_value:
            ordered.insert(0, [pop_size, val])
        else:
            ordered.append([pop_size, val])

    return map(list, zip(*ordered))
```

---

## A.3 Fine-tuning Parameters

### A.3.1 ERC

Applied to ERC on EmoryNLP, models were trained using an Adam optimizer, batch size of 32, $l2$ weight of 0.01, and learning rate of $2e - 5$. For MELD, models were trained for 10 epochs and 120 warmup iterations, while for EmoryNLP models were trained for 12 epochs with 160 warmup iterations.

### A.3.2 Player Performance Projection

To project player performance, simple, linear prediction heads were added on top of our pre-trained models and fine-tuned using an Adam optimizer with a learning rate of `2e-4` and *L2* of `1e-6` for up to eight epochs, minimizing the mean-squared-error between the model's predictions and ground-truth labels. The model's token-prediction linear head is replaced with a new *performance projection* head of shape $D_{model} \times 3$, as the model predicts three values.

### A.4 Prescribed vs Assigned Distribution

Figure 5 contains additional visualizations of the cluster assignments presented in Figure 3a of section 5.2.1 to better understand the differences between our cluster assignment mechanism and that of SoLar. We see that in the face of poor-quality predictions (low $\mu$), SoLar tends to drastically overestimate the most populous classes and underestimate the most populous ones. While our proposed method is not perfect, it alleviates this issue to a large extent.

### A.5 Additional LT-PLL Results

To allow for a direct comparison with RECORDS, we perform additional experiments on CIFAR100-LT with $\gamma = 100$ and $\phi = 0.03/0.05/0.07$, presenting the results in 7. As we can see, our approach improves upon their reported scores in all cases.

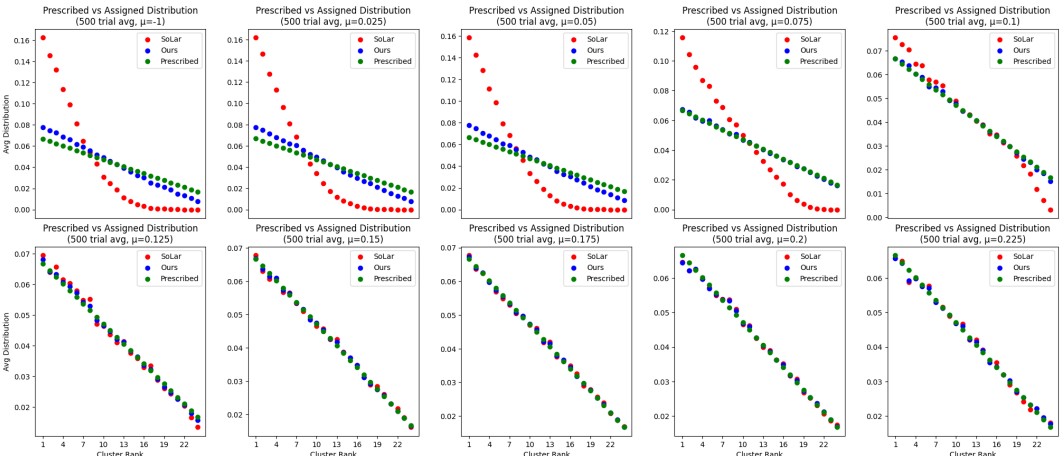

Figure 5: Visualization of cluster assignments resulting from ours/*SoLar* when presented with predictions of varying strength.

| Method | $\phi$=0.03 | | | | $\phi$=0.05 | | | | $\phi$=0.07 | | | |
|---|---|---|---|---|---|---|---|---|---|---|---|---|
| | Many | Medium | Few | All | Many | Medium | Few | All | Many | Medium | Few | All |
| CORR+ RECORDS | 66.37 | 42.54 | 13.77 | 42.25 | 68.49 | 40.20 | 8.50 | 40.59 | 69.97 | 36.71 | 4.37 | 38.65 |
| OURS+ RECORDS | 74.19 ±0.62 | 46.83 ±0.56 | 14.89 ±1.78 | 45.32 ±0.87 | 72.88 ±0.68 | 45.60 ±1.46 | 12.53 ±1.10 | 43.69 ±0.83 | 72.09 ±0.52 | 42.79 ±1.48 | 11.04 ±1.33 | 41.98 ±0.91 |

Table 7: Additional results on CIFAR100-LT with $\phi = 0.03/0.05/0.07$ and $\gamma = 100$.

