# OpenReview forum: "Clustering Entity Specific Embeddings Towards a Prescribed Distribution"
_ICLR.cc/2024/Conference — Submitted to ICLR 2024_

### Official Review · Reviewer_Sd3V · 2023-10-31

**Soundness:** 3 good
**Presentation:** 2 fair
**Contribution:** 3 good
**Rating:** 6
**Confidence:** 3

**Summary:**

Multi-entity sequences are common in the realm of NLP. However, existing approaches towards deriving entity-specific embeddings are typically only able to construct embeddings for a single entity at a time. The paper proposes a general method for deriving entity-specific embeddings from a multi-entity sequence completely within the transformer architecture. The method showcases SOTA performance on emotion recognition in MELD and EmoryNLP datasets, and on player performance projection task. In addition, the paper proposes a novel distribution-aware clustering method in the absence of labels, which attains SOTA performance in long-tail partial-label learning.

**Strengths:**

- The paper proposes a general method to extract entity-specific embeddings from a multi-entity sequence completely within the transformer architecture.

- The paper proposes a distribution-aware clustering method that achieves SOTA on the task of LT-PLL.

**Weaknesses:**

- For the experiments in Sec 5.1, it is unclear whether the baseline model has undergone similar “pre-training” in the target (i.e., REC/sports) domain. This casts doubt on whether the performance gain could be attributed to the “pre-training” on the target domain.

- The introduction of the player performance projection task, given its relative unfamiliarity, necessitates further clarification. It would be better if the authors could provide some examples of the task, especially in relation to the extraction of entity-specific embeddings from multi-entity sequences.

- The paper concurrently presents two distinct methodologies: one centered on extracting entity-specific embeddings and the other on a distribution-aware clustering approach. Presenting these two methods in the same paper seems unconventional, as it appears the two approaches are tackling different challenges and are not interdependent.

**Questions:**

- In Table 4, while EE+E2E ensembled with the SPCL-CL-ERC outperforms SPCL-CL_ERC, the performance of EE+E2E is lower than SPCL-CL-ERC. It is unclear why the performance of EE+E2E & SPCL-CL-ERC can be used to show that the proposed method achieved SOTA on MELD. Could you elaborate on the ensemble techniques utilized when combining EE+E2E with the SPCL-CL-ERC?

---

> ### Author Response · Authors · 2023-11-13
>
> The authors would like to thank you for the specific feedback provided. We appreciate the details that we can address either by explaining below or in the camera-ready version.
> * Yes, baseline methods presented in 5.1 - we believe you are specifically asking about Tables 2, 3, and 5 - were subjected to pre-training on the same datasets as our proposed models. The only difference is that they did not leverage our proposed approach for deriving entity-specific embeddings. We are happy to change the header for the third column of Table 2 from ‘Fine-tune’ to ‘Extended Pre-training’ if this would make things more clear.
> * Traditional sports analytics methods do not view the game of baseball as a sequence, but as a set of summary counting statistics, reminiscent of the “Bag of Words” approach from the early days of computational statistics. That is, they simply count the number of times each entity induced a certain event. These vectors of counting statistics are then used as input to some machine learning model - XGBoost/DNN/etc. We are trying to effectuate a change in the way sports analysis is done. Examples of the task as given in the paper are as follows:
>    * “Specifically, the model predicts how many strikeouts (K), hits (H), and walks (BB) starting pitchers will record against the opposing team’s starting batters and vice-versa”
> In baseball, a starting pitcher starts and pitches until substituted by a relieving pitcher. We seek to predict the number of times the pitcher will strikeout (three strikes to the same batter) opposing batters, number of hits he is likely to give to these batters, and the number of times he will pitch four balls (as opposed to strikes to the same batter) aka walks to batters. This prediction is done at the start of the game using the prior history (sequence) in the last 15 games. In each game, there are nine batters and one pitcher. We isolate the impact each individual batters have from that sequence and predict the number of strikeouts/hits/walks the batter will have in a game.Each entity is a player. Multi-entity refers to 9 players - batters in a team. The sequence here is a sequence of in-game events, such as hits, walks, runs, outs, etc., which constitute a baseball game. Each event is influenced by one or more of the afore-mentioned entities, i.e., the 9 batters and the pitcher. Our model processes the sequences once for all the players instead of having to iterate and learn the information about each player separately.
> * As mentioned above, we will explain better how these two are linked in the camera-ready version. The two methods are quite related to each other. One method enables a transformer-based model to derive entity-specific embeddings from a multi-entity sequence, and the other provides a means for guiding these constructed embeddings towards a prescribed distribution. The vanilla entity-specific embeddings provide utility on their own, but additional signal can be teased out by clustering them towards some prescribed distribution. Particularly in the domain of sport, the clustering of player embeddings towards a prescribed distribution cannot be performed without our proposed method for entity-specific embeddings - we are not aware of a method that allows the derivation of entity-specific embeddings from multi-entity sequences completely within the transformer. That said, each method can be utilized independently, as we demonstrate in our experiments - utterance-specific (entity-specific) embeddings can derived for each utterance in a multi-turn conversation, and representations of images with different categorical labels can be clustered towards a prescribed distribution in the LT-PLL task. Our embeddings describe player skills but then we show using a down-stream application that when we cluster using these embeddings, we get clusters of players, who in real-life using our baseball knowledge, we know are similar with respect to their playing characteristics or skills (e.g., home run hitter, contact hitter, etc.). Similarly, in a conversation/dialog application, the embeddings capture the state of the dialog. Again, using these embeddings, we could potentially cluster similar speakers (persuasive speakers, irascible speakers, etc.). The fact that the clustering works well illustrates the value-add of our embedding scheme. This is why we presented these two together. We think that together they make our argument about the utility of our method. However, if requested, we are willing to remove or shorten the clustering experiments in the camera-ready version.
> * The fact that an ensemble of EE+E2E and SPCL-CL-ERC yields SOTA, i.e., a stronger performance than either approach in isolation demonstrates that they uncover different signals useful in classifying the utterance. If they captured the same information, then an ensemble would not out-score either individual model. The results demonstrate that our approach uncovers signals not leveraged by SPCL-CL-ERC.

---

### Official Review · Reviewer_k1pV · 2023-11-01

**Soundness:** 2 fair
**Presentation:** 1 poor
**Contribution:** 2 fair
**Rating:** 3
**Confidence:** 4

**Summary:**

The paper introduces a method for deriving entity-specific embeddings, in which an entity can represent a speaker in dialogues or a player in sport analytics. The authors propose a method to automatically identify entities to be derived, in which each entity can interact with input tokens and other entities.  In addition, they propose a deep clustering method for embedding clustering, which modifies “Sinkhorn Knopp algorithm” by adjusting the probability mass of each cluster.

**Strengths:**

1. The proposed deep clustering method shows good performance in long-tail partial-label learning.
2. The paper explores sport analytics and long-tail partial label problems, which are both interesting problems.

**Weaknesses:**

1. The paper's structure and presentation need improvement, with concepts scattered and a lack of formal details. Clarity could be enhanced with a graphical overview. I would also suggest separating preliminaries from related work so that readers can better focus on what they need to learn before the method section. The details of the proposed method such as the formal formula for deriving entity specific embeddings are missing. In addition, there are many typos or unclear sentences which makes the paper hard to understand.
2. The paper introduces two methods (i.e. deriving entity specific embedding and clustering) that are not highly related to each other. The weak connection of the methods makes the paper difficult to understand.  The motivation for clustering may need clarification to avoid the perception of merging two separate papers.
3. The idea of ​​deriving entity-specific embeddings has also been explored when learning personalized embeddings for chatbots. This line of work can be discussed or compared in the paper.

**Questions:**

Please refer to the Weaknesses.

---

> ### Author Response · Authors · 2023-11-13
>
> With all due respect, we do not “propose a method to automatically identify entities to be derived”. In the second paragraph of Section 3, we clearly state the process of identifying entities for which embeddings should be derived “is often as simple as identifying the speaker or utterance ID associated with each timestep of the sequence.” Each dataset denotes the entities present in each record.
> * With respect to a graphical overview, Figure 2 contains a visual depiction of the key elements of the method for deriving entity-specific embeddings - entity embeddings are introduced at the beginning of the sequence, and the attention mask is updated such that each entity embedding can only tend to portions of the sequence with which the corresponding entity interacts. If by graphical overview, the esteemed reviewer meant something else, we would like to know as to what he/she would like that graphical overview to look like differently than Fig. 2.
> * We are of course happy to rectify any typos for a camera-ready version but we, unfortunately, did not find these many typos. We have written the paper as clearly as we could and would appreciate pointers to which sentences are unclear or contain typos.
> * We propose a method. Typical embeddings do not have closed-form formulas but a sequence of steps describing the procedure to build them. We modify the standard transformer model to handle our problem and state clearly the adaptations we have made in the paper. The entity embeddings are processed as any other item in the sequence would be. Our novelty lies in the way in which the model can 1) derive embeddings for individual entities (subsequences of interest) in a multi-entity sequence, and 2) leverage the relations between entities in a given record, all within the transformer architecture.
> * The two proposed methods are very much related to one another - the first method provides the tools to derive entity-specific embeddings from multi-entity sequences, and the second guides said embeddings towards a particular distribution. While the vanilla entity embeddings provide useful signals towards player performance projection (and the NLP task of emotion recognition in conversation), the method for distribution-aware clustering helps elicit certain qualities in the entity embeddings. Having said that, we understand that despite our best efforts, the motivation and the connection between the two sections were not clear. We will include the explanations given here and in response to the esteemed reviewer above in our camera-ready version to make the link crystal clear.
> * We are not sure which specific works you are referencing for “personalized embeddings for chatbots” but can respond generally. We would appreciate brief exact pointers (DOIs will do) to what the reviewer has in mind. Trying to guess, two topics come to our mind: 1) providing a LLM agent with a “persona” prompt, describing the general manner in which the agent should interact, and 2) leveraging some historical responses by the user to implicitly derive an embedding for the user that can be used to guide/improve agent responses. In both cases, the embedding is used to elicit a certain behavior from the LLM agent, which is not the objective of our work which seeks to “make sense” of things that already happened. In that sense, our work is most similar to the first part of the second topic - making sense of previous user responses. The major difference, however, is that methods for “personalized chat bots” typically only analyze the historical responses of an individual user - the current user interacting with the agent. That is, the model is presented with a long sequence that is the concatenation of all responses from an individual user. Since the responses are only from one user, it is not a multi-entity sequence. While current LLM chatbots only interact with an individual user at a time, forgoing the need to process multi-entity sequences, the ability to process multi-entity sequences is fundamental for other applications, such as in the domain of sport. That said, we will discuss this in the camera-ready paper. If provided a pointer to the papers the reviewer had in mind, we will compare and contrast against those in the camera-ready version.

---

### Official Review · Reviewer_hHfi · 2023-11-01

**Soundness:** 3 good
**Presentation:** 3 good
**Contribution:** 3 good
**Rating:** 5
**Confidence:** 3

**Summary:**

The authors propose a general method for deriving entity-specific embeddings completely within the transformer architecture and demonstrate SOTA results in the domains of natural language processing and sports analytics.
The proposed method first identifies the entities and then adds the entity embeddings to the input sequence. The added entity embeddings can only tend to indices in which the corresponding entity participates.
The authors further improve the Sinkhorn Knopp algorithm for clustering.

**Strengths:**

1. The authors have done a broad experiment and performed better on almost all the datasets.
2. The proposed method is reasonable and can lead to better entity representations.
3. The ablation study shows that both the entity embedding method and the clustering method can lead to better performance.

**Weaknesses:**

1. Adding special tokens to get embedding seems trivial. It is one of the common tricks to get the representation of entities with multiple tokens, such as the method in "Empowering Language Models with Knowledge Graph Reasoning for Question Answering".
2. The proposed clustering method and the embedding method seem to be two orthogonal directions. Feel like merging two not-quite-novel methods together.

**Questions:**

I see InstructERC is based on the much larger Llama2 model (350M vs 7B parms). Is there any experiment based on LLM?

---

> ### Author Response · Authors · 2023-11-13
>
> * Learning special entity-specific embeddings has been explored previously, as in the paper you reference. The key difference is that previous approaches primarily learn a single embedding for each entity of interest based on how that entity has been used across all records in some (pre-)training dataset. In our approach, the entity-specific embedding is constructed based on how the entity influences an individual record. Each entity embedding is the same when presented to the transformer, but is updated differently in each layer of the transformer based on the portion of the input sequence with which the said entity interacts. This design choice was made specifically for application in the domains where the entities are baseball players or multiple conversation partners. In the domain of sport (conversation), Player A (Person A), for example, may perform very differently in April than they do in July (may discuss things differently after they know each other than when they first met), and a static player (person) embedding would not account for such differences. Our method for deriving entity-specific embeddings from a sequence describing Player A’s recent games (Person A’s recent utterances)  addresses this.
> * The method also improves performance on tasks that are not typically thought of as being entity-based, such as emotion recognition in conversation. Here, the entity is assumed to be each individual utterance. Here, the best method for learning an entity-specific (utterance-specific) embedding is non-obvious. The representation need not be dynamic - each utterance only appears once - but because each utterance only appears once, there is limited training data from which to derive an embedding. We would be happy to add language to the paper to make this more clear.
> * While the two proposed methods can certainly be used independently of each other - the distribution-aware clustering method could be applied to other types of embeddings, for example, as shown in the LT-PLL experiments - we do not view them as orthogonal to one another. One method presents a way for deriving an entity-specific embedding from a multi-entity sequence, and the other a method for guiding said embeddings towards some prescribed distribution thereby going hand in hand.
> * With respect to Llama2 and other LLMs: No, we do not perform any experiments using LLMs. To the best of our knowledge, all of the recently released LLMs are autoregressive. However, as it stands, our method can only be applied to bi-directional models. Specifically, the method needs to know which indices correspond to a particular entity of interest (utterance in this case) at each layer of the model. This is not the case for autoregressive models, however, as the input sequence at timestep t denotes the token present at timestep t. During processing, though, the embedding at timestep t is updated to describe what the model thinks the next token will be, i.e. t+1. Although adapting our method to work with LLMs is an interesting path forward, we would like to note that our method is very competitive with the Llama2-based InstructERC - each method outperforming the other in one of the two benchmarks. Although LLMs are proficient in a wide range of tasks, they may not be the apt model for every application.

---

### Official Review · Reviewer_JpxY · 2023-11-03

**Soundness:** 2 fair
**Presentation:** 1 poor
**Contribution:** 2 fair
**Rating:** 3
**Confidence:** 2

**Summary:**

This work focuses on embedding learning, specifically deriving entity-specific embeddings in a multi-entity sequence within transformer architecture models. They address the limitations of the prior art in computing entity-specific embeddings when given a multi-entity sequence within transformer architecture and avoiding additional postprocessing outside transformer and redundant computations. The authors also present a method for distribution-aware clustering in the absence of labels, again addressing the deep clustering challenges of prior art that are dependent on labels.
The proposed approaches of learning entity-specific representation are applied to sport domain data and have shown improved performance.

**Strengths:**

+ the paper in incremental in the sense that extends the prior art in computing entity-specific embeddings when given the multi-entity sequence
+ the paper has also advanced the prior art: Sinkhorn-knopp algorithm for cluster assignments for example to identify clusters of players who impact the game in similar fashion and thereofre, further extend the distribution aware clustering to compute representations in absence of labels
+ the proposed approach has shown improvements on datasets from sports domains

**Weaknesses:**

- the paper is not an easy read and not easy to follow (too dense)
- the presentation of concepts is too scattered and missing a natural flow of reading, with full of abtract concepts and citation to prior art
- missing illustrations of the motivation and the problem statement
- the work is incremental and lacks novelty (or limited by the presentation of the paper), extending the transformers

**Questions:**

-

---

> ### Author Response · Authors · 2023-11-13
>
> We would like to point out that the proposed methods have been applied to the three domains - NLP, CV, and sports analytics. Only sport is listed throughout the review.
> * We have organized the paper to the best of our ability. First, we introduce a method for deriving entity-specific embeddings, and then a method for clustering said entities towards a prescribed distribution. With all due respect, we truly fail to see where we have introduced “full of abstract concepts”. We request the reviewer to kindly point these out. We extend the transformer architecture used in NLP and beyond to better handle multi-entity sequences, which we believe is necessary in various applications ranging from NLP to baseball.
> * With regards to “full of … citation to prior art”, we believe that we should cite prior art fully as scientists and have tried to do so.
> * With respect to flow, individual readers have different preferences and we used what we believe would be the best flow to illustrate the points we are making. We would again request the reviewer to point out where our flow is unnatural.
> * We would be grateful if you can please point out what you would like illustrated. Again, we believe we have illustrated our methods to the best of our abilities and would like to know how one can make this better. For example, Figure 2 describes the key elements of the entity-embedding process - the introduction of new entity embeddings at the beginning of the sequence and the modifications to the vanilla attention mask.
> * With respect to our motivation we state in several places clearly:
>    1. “... when processing a multi-speaker dialogue, for example, the [CLS] will describe the entire dialogue not a particular utterance. Existing approaches to address this often either involve redundant computation or non-trivial post-processing outside of the transformer.”
>    2. “While directly useful for performing many downstream tasks, the processed [CLS] embedding provides less utility for entity-specific tasks when given a multi-entity sequence. Consider the task of utterance-level emotion recognition in dialogue - the [CLS] embedding describes the dialogue, not any particular utterance.”
>    3. “Existing approaches towards deriving entity-specific embeddings are typically only able to construct embeddings for a single entity at a time, incurring redundant computation, or require non-trivial post-processing outside of the transformer architecture.”
>    4. “To this end, we propose a novel, general approach for the derivation of entity-specific embeddings from a multi-entity sequence and apply the proposed method to the domains of NLP and sports analytics. By equipping a transformer-based model with our entity-specific tooling, we demonstrate appreciable gains in both the pre-training and fine-tuning phases of learning. In the domain of NLP, we demonstrate how our approach can be used to achieve SOTA results for the task of emotion recognition in conversation (ERC) on the MELD and EmoryNLP datasets. Furthermore, in the domain of sport, our approach outperforms current statistical approaches and a transformer-baseline.”
> * We are a tad confused as to  how incremental improvements over prior work can be both a positive and a negative. While we strive for the best results we can, it is uncommon for a single paper to present results for various domains that are leaps and bounds greater than the previous SOTA. Without trying to elevate this work to the introduction of the transformer architecture, consider the original ‘Attention is all you need paper’ where the primary results were for translation. The transformer yielded improvements of 7.73%/1.24% over previous SOTA for EN-DE/EN-FR. Our method improved masked language modeling perplexity scores by 2.59%/71.55% in the domain of language/sport, weighted-F1 by 1.61%/13.41% on the emotion recognition benchmarks of MELD/EmoryNLP, R2 by 14.28% for player performance projection in MLB, and accuracy an average of 5.33% on two different LT-PLL datasets. Furthermore, our approach is a paradigm change for baseball analytics - previous methods view baseball as a set of summary counting statistics, akin to the bag-of-words approach from the early days of computational linguistics - described in Section 2.3.
> * To the best of our knowledge, our method has not been proposed by any other existing works. Again, we would request the reviewer to kindly point us to works that have proposed how one can (a)  handle multi-entity sequences in a principled manner, (b) reduce the redundancy involved in say handling “n” utterances using “n” separate passes, instead, using one pass to process “n” sequences thereby reducing computation, and (c) makes significant performance gains as our technique has made across multiple diverse domains. Application in this unexplored domain of sport requires modifications to prior work and such modifications can be used to advance SOTA in more traditional realms of DL (NLP/CV).

---

### Meta-Review · Area_Chair_cfmx · 2023-12-12

**Metareview:**

The paper presents an improved approach for extracting embeddings specific to entities and clustering them to align with a predefined distribution. While the reviewers largely agree that there are merits in this work, common concerns have been raised. One of the most significant issues is that the paper proposes two approaches to address different challenges. This aspect makes the work less compelling, as it appears to be a combination of two efforts (without very novel solutions). Other concerns include clarity and presentation issues, as well as incomplete comparisons with prior work, among others.

**Justification For Why Not Higher Score:**

There are strong concerns from multiple reviewers.

**Justification For Why Not Lower Score:**

N/A

---

### Decision · Program_Chairs · 2024-01-16

Reject